# Mechanisms of organelle biogenesis govern stochastic fluctuations in organelle abundance

**Shankar Mukherji[1,2,3], Erin K O'Shea[1,2,3,4]***

[1]FAS Center for Systems Biology, Harvard University, Cambridge, United States; [2]Department of Molecular and Cellular Biology, Harvard University, Cambridge, United States; [3]Howard Hughes Medical Institute, Harvard University, Cambridge, United States; [4]Department of Chemistry and Chemical Biology, Harvard University, Cambridge, United States

**Abstract** Fluctuations in organelle abundance can profoundly limit the precision of cell biological processes from secretion to metabolism. We modeled the dynamics of organelle biogenesis and predicted that organelle abundance fluctuations depend strongly on the specific mechanisms that increase or decrease the number of a given organelle. Our model exactly predicts the size of experimentally measured Golgi apparatus and vacuole abundance fluctuations, suggesting that cells tolerate the maximum level of variability generated by the Golgi and vacuole biogenesis pathways. We observe large increases in peroxisome abundance fluctuations when cells are transferred from glucose-rich to fatty acid-rich environments. These increased fluctuations are significantly diminished in mutants lacking peroxisome fission factors, leading us to infer that peroxisome biogenesis switches from de novo synthesis to primarily fission. Our work provides a general framework for exploring stochastic organelle biogenesis and using fluctuations to quantitatively unravel the biophysical pathways that control the abundance of subcellular structures.

**\*For correspondence:** Erin_Oshea@harvard.edu

**Reviewing editor**: Vivek Malhotra, Center for Genomic Regulation (CRG), Spain

## Introduction

Stochasticity in the abundance of cellular components is an intrinsic feature of biological systems. But while noise in molecular-scale processes such as gene expression and signal transduction have been examined in great detail, fluctuations have not been systematically characterized at one of the most critical scales of biological organization: compartmentalization of the eukaryotic cell into organelles. Two fundamental questions in systems cell biology are: how precisely does the cell control the number of a given organelle, and how do the mechanisms underlying organelle biogenesis affect this precision? It has been suggested, for example, that the cell may be able to count and tightly control the number of a given organelle (*Rafelski and Marshall, 2008*). However, we lack comparisons of theoretical calculations of cell-to-cell variability in organelle abundances to experimentally measured organelle abundance distributions.

There are four basic processes resulting in organelle abundance changes that can give rise to fluctuations in organelle number: de novo synthesis from a pre-existing membrane source, fission (*Lowe and Barr, 2007*), fusion (*Denesvre and Malhotra, 1996*), and decay such as through partitioning during cell division or autophagy (*van der Vaart et al., 2008*). In the budding yeast *Saccharomyces cerevisiae*, Golgi abundance is the result of a steady-state determined by the balance of de novo synthesis and decay through maturation and dilution by cell division (*Rossanese et al., 1999*; *Bevis et al., 2002*; *Losev et al., 2006*; *Matsuura-Tokita et al., 2006*); though modest levels of Golgi fusion have been reported (*Bhave et al., 2014*) decay appears to be dominated by maturation (*Losev et al., 2006*). Vacuolar abundance is thought to be primarily determined by the balance of fission and fusion

**eLife digest** Any cell that has a nucleus also contains various other organelles, such as the mitochondria that generate energy inside the cells. Like the nucleus, most of these organelles are enclosed within a membrane. Unlike the nucleus, however, there can be two or more copies of other types of organelles in a healthy cell.

How do the numbers of the different organelles in a cell change? The copy number for a given organelle can be increased in two ways: by the synthesis of new organelles, or the fission of an existing organelle to form two new organelles. Conversely, the number of organelles can also be decreased in two ways: an organelle can decay, or two organelles can fuse to form one new organelle. The steady state for a given organelle results from a balance of these creative and destructive processes.

Researchers have thought for some time that cells are able to count how many organelles of a given type they contain. It was also thought that cells have some control over this number, but it was not known how precisely cells could control the number of organelles they contained. It was also not known how this level of precision was influenced by the different processes responsible for making new organelles.

To address these issues, Mukherji and O'Shea have developed a stochastic model that treats the processes of organelle creation and destruction as if they were simple chemical reactions. A tool from statistical physics, known as the fluctuation-dissipation theorem, was then used to analyze this model and derive an equation that predicts how the fluctuations in organelle number depend on the rates of the processes that govern organelle number.

Mukherji and O'Shea used this model to make predictions about various organelles in the budding yeast *S. cerevisiae*. For two of these—vacuoles and the Golgi apparatus—the processes that lead to an increase or decrease in the number of organelles are well understood. In both cases the model accurately predicted the level of fluctuations measured in experiments. Moreover, Mukherji and O'Shea found that cells exhibited the maximum predicted level of fluctuations that could be generated by the processes that either increased or decreased the number of each organelle.

The model was also able to shed light on a long-running debate over the cellular origins of an organelle called the peroxisome. This organelle—which is involved in breaking down fatty acids and other compounds—has been studied much less than the Golgi apparatus and vacuoles, but there is compelling evidence that new peroxisomes are created by de novo synthesis and by the fission of existing peroxisomes.

Mukherji and O'Shea found that fluctuations in the number of peroxisomes suggest that the production of new peroxisomes is dominated by fission when the yeast cells are grown in a medium that is rich in oleic acid: peroxisomes are metabolically active and proliferate rapidly in such a medium. In a glucose-rich medium, on the other hand, most new peroxisomes are produced by de novo synthesis. The case of the peroxisome thus highlights the possibility of extending this mathematical framework to explain the creation and destruction of organelles and other subcellular structures in a range of organisms and environments.

events (*Wickner, 2002*), though vacuoles can be generated de novo in mutant yeast strains with impaired vacuole inheritance pathways (*Banta et al., 1988*; *Catlett and Weisman, 2000*) and vacuolar membrane is generated de novo in cells to support vacuolar growth; the quantitative contribution of de novo vacuole biogenesis to vacuole copy number remains uncharacterized in physiological conditions. For other endomembrane organelles, however, the processes underlying abundance changes are even less clear. Peroxisomes, for example, are thought to increase in number by both de novo synthesis (*Hoepfner et al., 2005*; *van der Zand et al., 2012*) and fission pathways (*Hoepfner et al., 2001*; *Yan et al., 2005*; *Motley and Hettema, 2007*). Mature peroxisomes do not undergo fusion events (*Motley and Hettema, 2007*). Furthermore, peroxisome abundances can be actively upregulated by culturing budding yeast cells in medium containing long chain fatty acids such as oleic acid (*Mast et al., 2010*). We lack a quantitative understanding of whether de novo synthesis or fission dominates the generation of peroxisomes in either steady growth in glucose or during active proliferation in oleic acid.

Here we present a stochastic model of organelle production, and combine a statistical property derived from the model, the Fano factor, with simple, experimentally-verified assumptions to yield two results. First, for organelles whose abundance-changing processes were previously characterized—the Golgi apparatus and vacuole—we made quantitatively accurate predictions about the size of organelle abundance fluctuations, leading to the surprising conclusion that the cell tolerates the maximum level of variability in Golgi apparatus and vacuole abundances generated by their biogenesis pathways. Second, for organelles where competing processes could contribute to organelle production but whose relative quantitative importance was unknown—the peroxisome—we used the model and experiment to infer that budding yeast cells switch from de novo synthesis dominated peroxisome production when grown in glucose-containing medium to fission dominated production in oleic acid-containing medium. Our theory of organelle biogenesis is very simple, but despite its simplicity we can gain mechanistic insight into the processes underlying organelle biogenesis. Though our framework suppresses details that could be relevant to organelle copy number regulation, such as when organelles share and thus compete for common biogenesis factors—as in the case of mitochondria and peroxisomes, which share fission factors in common for example—or when different processes affecting organelle copy number interact, such as the Golgi checkpoint regulating the progression of the mammalian cell cycle in which organelle biogenesis and decay would be coupled (*Sutterlin, et al., 2002*), it is easily extendable to account for such effects. We therefore anticipate it will be a useful framework in which to analyze the pathways underlying organelle creation and destruction and subcellular structures more generally.

## Results

### Simulations reveal distinct effects of organelle biogenesis mechanisms on fluctuations in organelle abundance

To evaluate the relative importance of different biophysical pathways in shaping organelle abundances, we constructed a mathematical model of organelle abundance dynamics (*Figure 1A*). As the average number of organelles is typically small, we formulated a model consisting of four coupled stochastic processes, each parameterized by an associated rate constant:

(i) de novo synthesis, in which new organelles are generated at a constant rate $k_{de\ novo}$ per time:

$$n \xrightarrow{k_{de\ novo}} n+1$$

(ii) fission, in which new organelles are generated at a rate $k_{fission}$ per organelle per time:

$$n \xrightarrow{k_{fission}n} n+1$$

(iii) homotypic fusion, in which organelles are destroyed at a rate $k_{fusion}$ per organelle squared:

$$n \xrightarrow{k_{fusion}n(n-1)} n-1$$

(iv) decay, representing an aggregation of a number of processes such as cell division, maturation, heterotypic fusion or autophagy, in which organelles are destroyed at a rate $\gamma$ per organelle per time:

$$n \xrightarrow{\gamma n} n+1$$

These stochastic processes are then aggregated to yield the following master equation that describes the dynamics of organelle biogenesis and decay:

$$\frac{dp_n(t)}{dt} = \left(k_{de\ novo} + k_{fission}(n-1)\right)p_{n-1}(t) + \left(\gamma + k_{fusion}n\right)(n+1)p_{n+1}(t)$$
$$- \left(k_{de\ novo} + k_{fission}n + k_{fusion}n(n-1) + \gamma n\right)p_n(t)$$

We performed stochastic simulations using the Gillespie algorithm (*Gillespie, 1977*) in which de novo synthesis (*Figure 1B*), fission (*Figure 1C*), and fusion (*Figure 1D*) were allowed to dominate over other abundance-changing processes. We then built organelle abundance histograms from 10,000 repetitions of the stochastic simulations, which reveal that each process leaves a distinct statistical

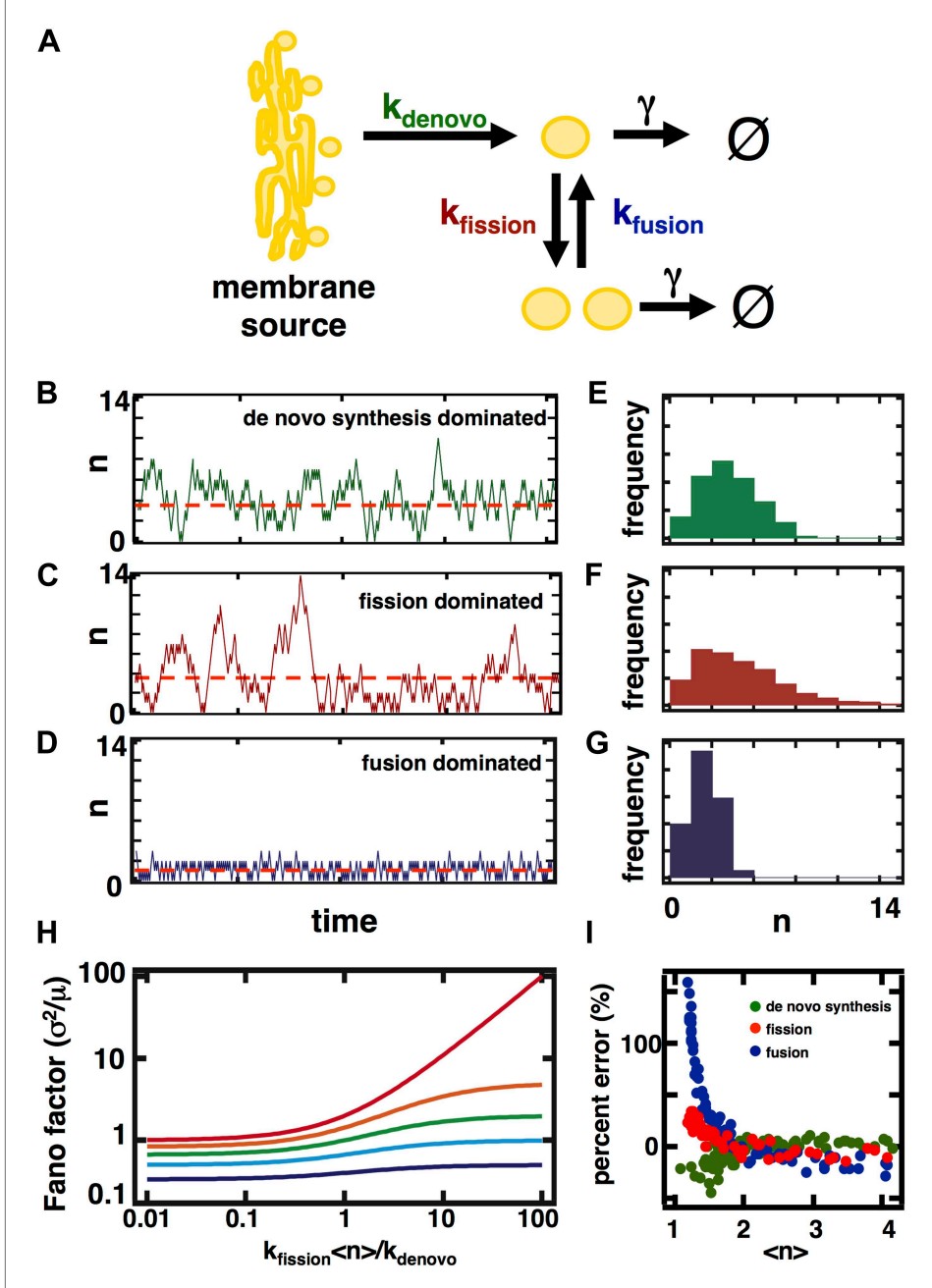

**Figure 1**. A stochastic model of organelle abundance dynamics. (**A**) Organelle abundances are governed by four distinct biophysical processes: (i) de novo synthesis ($k_{de\ novo}$, with units per time); (ii) fission ($k_{fission}$, with units per organelle per time); (iii) fusion ($k_{fusion}$, with units per organelle squared per time); and (iv) decay ($\gamma$, with units per organelle per time). (**B**) Sample trajectory generated by a Gillespie simulation of the model depicted in (**A**) in which the parameters are chosen to allow de novo synthesis to dominate over fission, fusion, and decay. (**C**) Sample trajectory as in (**B**), but with parameters chosen to allow fission to dominate over de novo synthesis, fusion, and decay. (**D**) Sample trajectory as in (**B**), but with parameters chosen to allow fusion to dominated over de novo synthesis, fission, and decay. (**E**) Histogram of the number of organelles generated by Gillespie simulations with parameters chosen as in (**B**). (**F**) Histogram of number of organelles generated by Gillespie simulations with parameters chosen as in (**C**). (**G**) Histogram of number of organelles generated by Gillespie simulations with parameters chosen as in (**D**). (**H**) Analytical approximation of the variance of the organelle abundance distribution divided by the mean of the distribution, or Fano factor, as a function of the ratio of the fission to de novo synthesis rate for various values of the fusion rate; from top to bottom the curves are calculated in order of

*Figure 1. Continued on next page*

*Figure 1. Continued*

increasing fusion rates: 0 (red), 0.25 (orange), 1 (green), 10 (cyan), 100 (blue). The value of γ is selected such that the mean organelle abundance is held constant for every point along these curves. (**I**) Percent error in the Fano factor prediction from the analytical approximation compared to Fano factor calculated from Gillespie simulations of the reaction scheme. The percent error is computed as the de novo synthesis (green dots), fission (red dots), and fusion (blue dots) rate constants are varied to tune the mean organelle abundance from <n> ≈ 1 to <n> ≈ 4, with the percent error between the prediction and simulation reaching close to 0% at <n> ≈ 2.
The following figure supplements are available for figure 1:

**Figure supplement 1**. Comparison of Fano factors calculated from simulations of model with first order decay process (Model 1) with two alternative models substituting first order decay with partitioning at regular intervals.

signature on the organelle abundance distribution (*Figure 1E–G*)—each process strongly affects the variance of the organelle abundance distribution. It is worth noting that our results are virtually unchanged if we incorporate processes mimicking partitioning of organelles at cell division (*Figure 1—figure supplement 1*), whether through deterministic reduction of organelle copy number by half or by probabilistic binomial partitioning to daughter cells at regular time intervals. To compare the variances of the distributions from different simulation regimes, we calculated the variance of the distribution divided by the mean of the distribution, also known as the Fano factor. We chose to work with the Fano factor, rather than alternative metrics for fluctuations such as the coefficient of variation, because it equals 1 for a Poisson distribution, and so we can readily detect the deviations from Poisson statistics that arise from fission and fusion mediated processes by comparing the observed Fano factor to 1.

## Fluctuation dissipation theorem allows analytical calculation of the organelle abundance Fano factor

To understand how de novo synthesis, fission, fusion, and decay affect the Fano factor for the abundance distribution of a given organelle, we analytically calculated the Fano factor for the reaction scheme depicted in *Figure 1A*. Building on earlier work from the theory of stochastic processes, *Paulsson (2005)* showed that the Fano factor for a given reaction scheme can be approximately calculated according to a restatement of the fluctuation dissipation theorem:

$$\frac{\sigma^2}{\langle n \rangle} = \frac{\langle |\delta| \rangle}{C}$$

where $\langle |\delta| \rangle$ is the average step size for the different possible stochastic processes contained in the reaction scheme (i.e., the 'fluctuation' component), and C is the adjustment that the reaction fluxes make in response to a given amount of change in the abundance (i.e., the 'dissipation' component):

$$C = \frac{\Delta R^- / R^-}{\Delta n / n} - \frac{\Delta R^+ / R^+}{\Delta n / n}$$

where $R^-$ is the flux of organelle death events, $R^+$ is the flux of organelle birth events. We refer the reader to a complete derivation of this result that is presented in great detail by *Paulsson (2005)*. Substituting the birth and death flux terms into the fluctuation-dissipation theorem and noting that since each process, whether de novo synthesis, fission, fusion or first order decay, adds or subtracts 1 organelle at a time therefore $\langle |\delta| \rangle = 1$, we obtain:

$$\frac{\sigma^2}{\langle n \rangle} = \frac{1}{\dfrac{k_{fusion}\left(2\langle n \rangle - 1\right) + \gamma}{k_{fusion}\left(\langle n \rangle - 1\right) + \gamma} - \dfrac{k_{fission}\langle n \rangle}{k_{de\ novo} + k_{fission}\langle n \rangle}} \tag{1}$$

where $\sigma^2$ refers to the variance and <n> to the mean of the organelle abundance distribution. Consistent with our simulation results, the analytical expression for the Fano factor yields a quantity that increases with an increase in the ratio of the rates of fission and de novo synthesis, but decreases with an increase in the rate of fusion (*Figure 1H*). Since *Equation 1* is an approximation to the true

Fano factor, it is important to assess the range over which it is valid. In *Figure 1I*, we compare our approximation for the Fano factor based on the fluctuation-dissipation theorem (*Equation 1*) to Fano factors obtained by simulating the model using the Gillespie algorithm. Specifically, we vary either the de novo synthesis (green dots), fission (red dots) or fusion (blue dots) rate constant while holding the others constant and for each combination of rate constants: (1) calculate the Fano factor using *Equation 1* and mean organelle abundance; and (2) simulate the model using the Gillespie algorithm and calculate the Fano factor and mean organelle abundance from the organelle abundance distribution generated by the simulation. We then define the 'percent error' to be:

$$\text{percent error} = \frac{\text{calculated Fano factor} - \text{simulated Fano factor}}{\text{simulated Fano factor}}$$

To plot the curves presented in *Figure 1I*, we plotted the percent error metric as a function of the mean organelle abundance with each dot representing a set of rate constants. As shown in *Figure 1I*, no matter which rate constant is varied, if the mean organelle abundance is larger than 1 then the difference between the exact Fano factor obtained from simulation and the result of *Equation 1* virtually disappears. Furthermore, as shown below, in physiologically relevant regimes the fluctuation-dissipation based approximation is highly accurate and in some cases reduces to the Fano factor obtained by exactly solving the master equation.

*Equation 1* can be broken down into distinct cases for different organelles, depending on which processes among de novo synthesis, fission, fusion and decay govern the abundance of a given organelle. Three of these cases are relevant to the organelles under study.

## Fluctuations in Golgi abundance match predictions from Poisson statistics

In case 1, corresponding to the Golgi apparatus, $k_{fusion} = k_{fission} = 0$ as the Golgi apparatus is only affected by de novo synthesis and decay through maturation in budding yeast (*Rossanese et al., 1999*; *Bevis et al., 2002*; *Losev et al., 2006*; *Matsuura-Tokita et al., 2006*; *Figure 2A*). In this limit *Equation 1* reduces to $\sigma^2/\langle n \rangle = 1$, reflecting the fact that de novo synthesis and first order decay operating alone corresponds to a Poisson process; one of the hallmarks of the Poisson distribution is that its variance equals its mean, and our approximate equation for the Fano factor reduces exactly to this limit. Thus, we would expect that the Golgi abundance distribution should yield a Fano factor of 1. To test this prediction, we performed spinning disc confocal microscopy on a budding yeast strain expressing the monomeric red fluorescent protein (mRFP) fused to the Golgi localized marker protein Anp1 (*Huh et al., 2003*). Anp1-mRFP forms punctate spots (*Figure 2B*) marking the presence of individual Golgi, whose number we quantified in each cell to generate a Golgi abundance histogram from which we could calculate the Fano factor. We see excellent agreement between the theory and experiment, as the measured Golgi abundance distribution closely matches the Poisson distribution derived from the experimentally determined mean Golgi abundance (*Figure 2C*) and we measure a Fano factor $\sigma^2/\langle n \rangle = 1.0 \pm 0.1$ (*Figure 2D*, red bar), in agreement with the theoretically predicted $\sigma^2/\langle n \rangle = 1$ (*Figure 2D*, blue bar). Interestingly, when we repeat these measurements for the late Golgi by performing confocal microscopy on a budding yeast strain expressing the green fluorescent protein (GFP) fused to the late Golgi marker protein Sec7, we see virtually identical results with the late Golgi abundance distribution closely matching a Poisson distribution (*Figure 2—figure supplement 1A*) and thus yielding a measured Fano factor of $\sigma^2/\langle n \rangle = 1.0 \pm 0.1$ (*Figure 2—figure supplement 1B*). Furthermore, in order to reduce potentially confounding extrinsic sources of fluctuations due to variations in the phase of the cell cycle each cell in our population is in, we synchronized the cell cycle phases of the cells in our experiments by arresting them in S-phase of the cell cycle through treatment with 100 mM hydroxyurea. We see that synchronizing the cell cycle phases of the cells we examine by microscopy does not affect the Fano factors of the measured abundance distributions for the Golgi or late Golgi (*Figure 2—figure supplement 2A–D*).

## Vacuole abundance fluctuations confirm predicted sub-Poissonian Fano factor inherent to fission-fusion balance

In case 2, corresponding to the vacuole, $k_{de\ novo} = \gamma = 0$ as the vacuole is only affected by fission and fusion (*Figure 2E*). In this limit, *Equation 1* reduces to $\frac{\sigma^2}{\langle n \rangle} = 1 - \frac{1}{\langle n \rangle}$, where $\langle n \rangle$ is the mean number of vacuoles. Given an experimentally measured mean number of vacuoles of 2.1 (*Wickner, 2002*; 'Materials and methods'), we make the non-trivial prediction that $\sigma^2/\langle n \rangle = 0.5$ for the vacuole

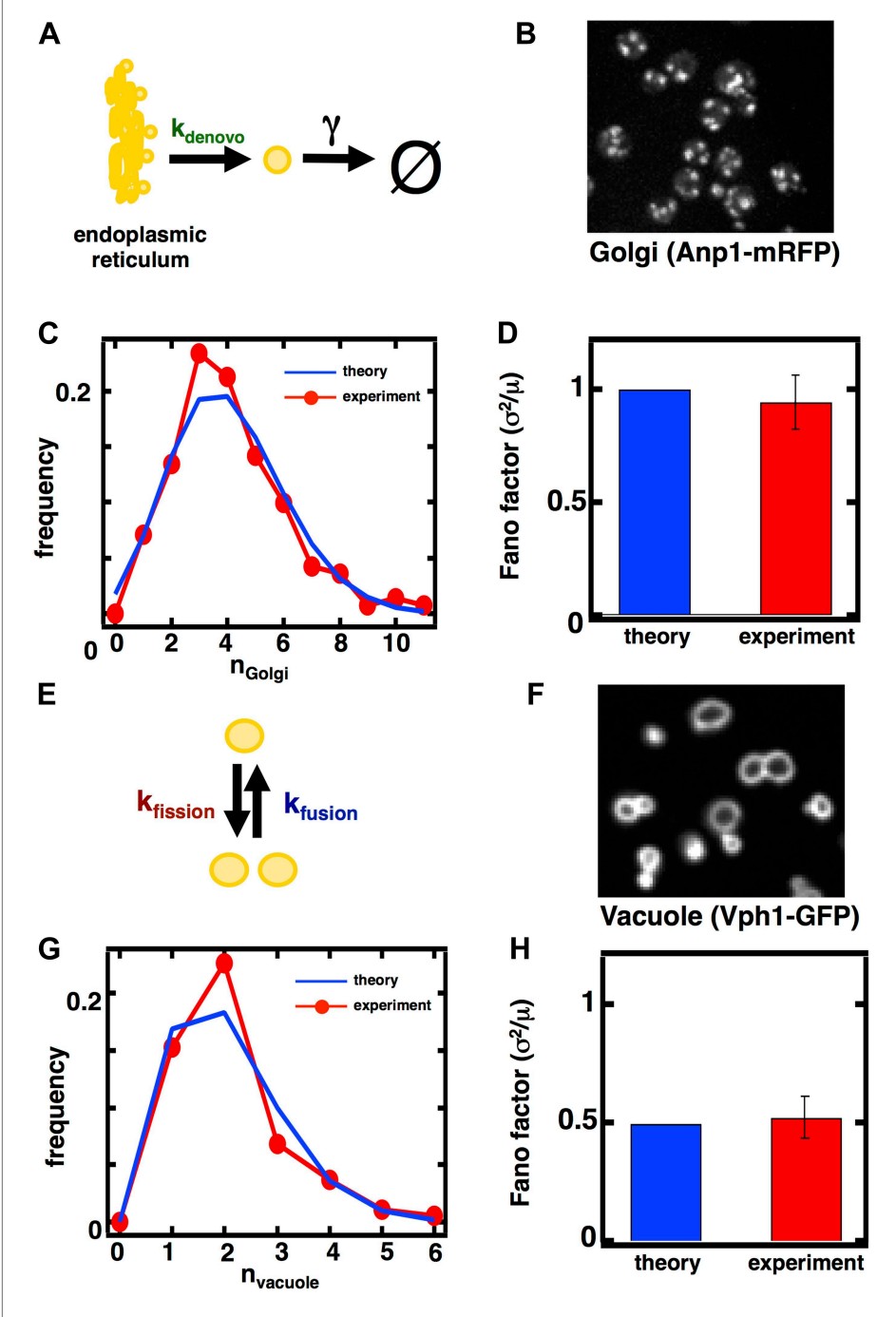

**Figure 2**. Predicting the stochastic fluctuations in Golgi apparatus and vacuole abundances. (**A**) Schematic depicting the biophysical processes that govern Golgi apparatus abundances. (**B**) Spinning disc confocal microscopy images of the Golgi apparatus as visualized by the fusion protein Anp1-mRFP. (**C**) Histograms depicting the theoretically predicted Golgi apparatus abundance distribution (blue trace) and experimentally measured single haploid cell Golgi apparatus abundance distribution (red trace). N = 141 cells were analyzed. (**D**) Bar graph depicting theoretical prediction (blue bar) and experimental measurement (red bar) of the Golgi apparatus abundance distribution Fano factor. (**E**) Schematic depicting the biophysical processes that govern vacuole abundances. Error bars are ±1 standard error of the mean, estimated by bootstrapping. (**F**) Spinning disc confocal microscopy images of the vacuole as visualized by the fusion protein Vph1 fused to green fluorescent protein (Vph1-GFP). (**G**) Histograms depicting the theoretically predicted vacuole abundance distribution (blue trace) and
*Figure 2. Continued on next page*

*Figure 2. Continued*

experimentally measured single haploid cell vacuole abundance distribution (red trace). N = 95 cells were analyzed. (**H**) Bar graph depicting theoretical prediction (blue bar) and experimental measurement (red bar) of the vacuole abundance distribution Fano factor. Error bars are ±1 standard error of the mean, estimated by bootstrapping.

The following figure supplements are available for figure 2:

**Figure supplement 1**. Predicting the stochastic fluctuations in late Golgi abundances.

**Figure supplement 2**. Predicting stochastic fluctuations in Golgi, late Golgi and vacuole abundance distributions in budding yeast cells synchronized and arrested in S-phase of the cell cycle.

distribution. This is a surprising and strong prediction of the model because a Fano factor of less than 1 resulting from fission and fusion processes operating together is purely due to the low numbers of vacuoles present at steady state; if <n> were much larger than 1, then the Fano factor would approach 1 and be indistinguishable from case 1. This distinction between case 1 and case 2 is a reflection of the fact that vacuole abundance, the result of a balance between fission and fusion events, follows a shifted Poisson distribution. In a shifted Poisson distribution, the probability of having n = 0 vacuoles is 0 because a cell can only reduce its vacuole numbers through fusion events to n = 1 vacuole, at which point the fusion rate $k_{fusion}\, n(n-1) = 0$. The Fano factor for the shifted Poisson distribution can be calculated exactly and yields the same expression as *Equation 1* with $k_{de\ novo} = \gamma = 0$.

To test the predictions that vacuole abundances follow a shifted Poisson distribution with Fano factor $\frac{\sigma^2}{\langle n \rangle} = 1 - \frac{1}{\langle n \rangle} = 0.5$, we visualize vacuoles by imaging a budding yeast strain that expresses the vacuolar membrane protein Vph1 fused to the green fluorescent protein (GFP; *Huh et al., 2003*). Vph1-GFP forms discrete rings (*Figure 2F*) that we count in each cell to construct vacuole abundance distributions, as was done for the Golgi. We see an excellent match between theory and experiment; we observe a distribution overlapping with the theoretically predicted shifted Poisson distribution (*Figure 2G*) and measure a Fano factor $\sigma^2/<n> = 0.5 \pm 0.1$ (*Figure 2H*, red bar), in agreement with the theoretically predicted $\sigma^2/<n> = 0.5$ (*Figure 2H*, blue bar); we obtain the same match between theory and experiment in cells whose cell cycle phases have been synchronized to be in S-phase through hydroxyurea treatment (*Figure 2—figure supplement 2E–F*). It is important to note that the close match between theory and experiment here suggests that de novo vacuole biogenesis, which is observed only in mutant strains that specifically disable vacuole inheritance, appears not to play a quantitatively significant role in affecting vacuole abundance in wild-type yeast. In an alternative model that allows de novo vacuole biogenesis to occur, *Equation 1* reduces to $\frac{\sigma^2}{\langle n \rangle} = \cfrac{1}{\cfrac{2\langle n \rangle - 1}{\langle n \rangle - 1} - \cfrac{k_{fission}\langle n \rangle}{k_{de\ novo} + k_{fission}\langle n \rangle}}$; if we

substitute the experimentally measured mean vacuole abundance of ≈2 into this expression and if we set this expression equal to the experimentally measured vacuole abundance distribution Fano factor of $\sigma^2/<n> = 0.5$, then solving for $k_{de\ novo}$ yields the result that $k_{de\ novo} \approx 0$.

Taken together, the cases of the Golgi apparatus and vacuole fluctuations allow us to make two conclusions. First, budding yeast cells tolerate the maximum level of variability generated by the biogenesis pathways governing Golgi apparatus and vacuole abundance, evidenced by the fact that our model predicted the experimental data with high quantitative accuracy without invoking any feedback control mechanisms to control the number of organelles. Second, as expected from theory, different biogenesis mechanisms generate differing levels of abundance fluctuations. At low mean organelle copy numbers, organelles governed by fission and fusion (vacuole; *Figure 2H*) inherently exhibit smaller abundance fluctuations than organelles governed by de novo synthesis and decay (Golgi; *Figure 2D*). In the case of the vacuole, our fluctuation analysis also sheds light on the quantitative role played by de novo vacuole biogenesis (*Catlett and Weisman, 2000*). In particular, even though up to 50% of the vacuole membrane is generated de novo in a given cell (*Catlett and Weisman, 2000*), actual vacuole copy number is likely not strongly affected by de novo vacuole biogenesis, consistent with the most widely accepted model of vacuole biogenesis (*Wickner, 2002*). Thus we can use experimentally measured fluctuations in organelle abundance to make quantitative inferences about the relative contributions of different organelle biogenesis pathways for cases where the pathways are less understood.

## Fluctuation analysis allows inference of a switch from de novo synthesis to fission dominated biogenesis of peroxisomes in fatty acid rich environments

Given the success of the model in making predictions about the abundance distributions and magnitude of fluctuations in Golgi and vacuole abundances, in our last case we sought to use the model to infer mechanistic insight into organelle biogenesis. In case 3, corresponding to the peroxisome, fusion is thought to be negligible and *Equation 1* reduces to $\frac{\sigma^2}{\langle n \rangle} = 1 + \frac{k_{fission} \langle n \rangle}{k_{de\,novo}}$ (*Figure 3A*).

We can thus use the Fano factor to infer whether de novo synthesis or fission dominates the production of peroxisomes; this is a topic of active debate (*Hoepfner et al., 2005*; *Motley and Hettema, 2007*). Specifically, in the absence of fusion, a Fano factor significantly larger than 1 indicates that fission dominates over de novo synthesis in generating an increased number of organelles, while a Fano factor close to 1 indicates that de novo synthesis dominates over fission. Even if $k_{fusion} > 0$, a Fano factor larger than 1 still indicates that fission dominates over de novo synthesis. For example if fusion and first order decay reduces organelle numbers at similar rates, $\frac{\sigma^2}{\langle n \rangle} = \frac{2k_{de\,novo} + 3k_{fission} \langle n \rangle}{3k_{de\,novo} + k_{fission} \langle n \rangle}$ which only grows much larger than 1 when $k_{fission} <n> >> k_{de\,novo.}$ Furthermore, if de novo synthesis dominates over fission, we also expect to see that the organelle abundance distribution will closely match a Poisson distribution, as we observed for the Golgi apparatus, while if fission dominates then we expect a distribution broader than Poisson. Notably, peroxisomes are greatly upregulated in number when yeast cells are cultured in fatty acid-rich medium; therefore it is of interest to measure Fano factors for these organelle abundance distributions in both glucose and fatty-acid rich media.

It is important to note, in applying our theory to peroxisome biogenesis, that these predictions are completely insensitive to whether or not the details of de novo peroxisome biogenesis, which occurs through fusion of pre-peroxisomal vesicles that bud off from the ER (*van der Zand, et al., 2012*), are included in the model or not. The model we use to calculate Fano factors with (*Figure 3A*) uses a simplified peroxisome biogenesis process in which de novo peroxisome biogenesis proceeds as a single step of a mature peroxisome budding from the ER. In an alternative, more biologically detailed model (*Figure 3—figure supplement 1A*) we explicitly keep track of the two vesicle types, one bearing the peroxisomal membrane proteins (PMPs) Pex2 and Pex10 on their surfaces and the other the PMPs Pex13 and Pex14 on their surfaces, that fuse in order to form the peroxisome import machinery that then imports the enzymes that allow the peroxisome to carry out its metabolic functions (*van der Zand, et al., 2012*). To check that our simplification of de novo peroxisome biogenesis does not introduce significant quantitative errors, we compare the mature peroxisome abundance statistics for the two alternative de novo biogenesis models (one-step vs pre-peroxisomal vesicle fusion), isolated from any contributions from fission. We see that when we set the fission rate to 0 and tune the production rates in the two alternative de novo biogenesis models to produce the same mean peroxisome abundance (*Figure 3—figure supplement 1B*), the alternative models of de novo peroxisome biogenesis produce abundance distributions whose Fano factors, $\sigma^2/<n> = 1.0$, are virtually identical to each other and to the value we expected from the Poisson distribution (*Figure 3—figure supplement 1C*). From this result we conclude that our original simple, one-step de novo biogenesis pathway model is statistically equivalent to the more detailed model of de novo peroxisome biogenesis that explicitly tracks the creation and fusion dynamics of Pex2/10 and Pex13/14 containing vesicles.

To visualize peroxisomes we imaged budding yeast strains containing fusions of the yeast enhanced monomeric Citrine (yemCitrine) protein to the peroxisome targeting signal PTS1, consisting of the amino acids serine, lysine, and leucine at the C terminus of the protein (*Figure 3B*); we also repeated our peroxisome experiments using a budding yeast strain containing a fusion of the peroxisome membrane protein Pex3 with the monomeric red fluorescent protein (mRFP; *Figure 3—figure supplement 2A*). The PTS1 and Pex3 reporters gave virtually indistinguishable results. After measuring single cell histograms of peroxisomes (*Figure 3C*, *Figure 3—figure supplement 2B*), we calculate that the Fano factors for the peroxisome abundance distributions for cells cultured in glucose is $\sigma^2/<n> = 1.1 \pm 0.1$ (*Figure 3D*, *Figure 3—figure supplement 2C*), which is also the peroxisome abundance distribution Fano factor we obtain from yeast cells arrested in S-phase of the cell cycle (*Figure 3—figure supplement 3*), leading us to infer that the organelle is generated primarily by de novo synthesis in glucose. In cells cultured in oleic acid-containing medium, however, we see a marked increase in the size of the organelle

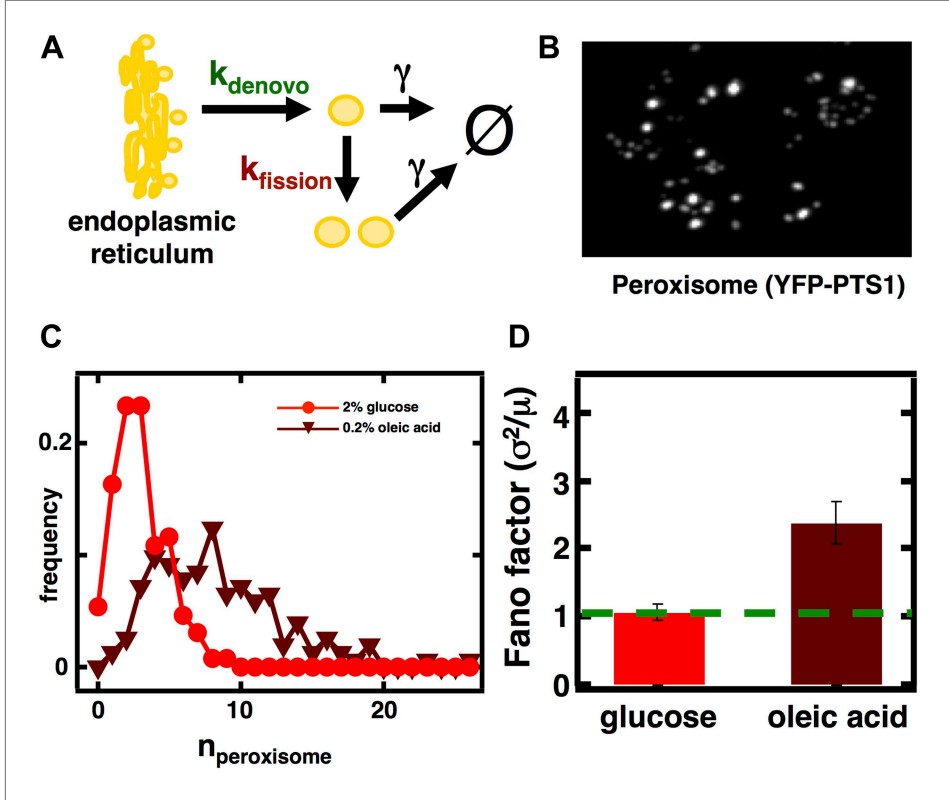

**Figure 3**. Inferring the dominant biophysical pathways in peroxisome biogenesis. (**A**) Schematic depicting the biophysical processes that govern peroxisome abundances. (**B**) Spinning disc confocal microscopy images of the peroxisome as visualized by the fusion protein YFP-PTS1-mRFP. (**C**) Histograms depicting experimentally measured single cell peroxisome abundance distributions for haploid cells grown in 2% glucose (red circles) and haploid cells grown in 0.2% oleic acid (dark red triangles). N = 129 cells were analyzed in glucose medium and N = 153 cells were analyzed in oleic acid medium. (**D**) Bar graph depicting measured peroxisome abundance distribution Fano factors in glucose-rich and 0.2% oleic acid-rich media. The green dashed line indicates a Fano factor $\sigma^2/<n> = 1$, marking the boundary between de novo synthesis and fission dominated organelle production. *Figure 3—figure supplement 1* depicts a peroxisome biogenesis model, referred to as Model 2, alternative to the model depicted in panel (**A**). *Figure 3—figure supplement 2* depicts data similar to panels (**B**–**D**) but with Pex3-mRFP as the peroxisome marker. *Figure 3—figure supplement 3* displays simulation results from Model 2 showing how increased pre-peroxisomal vesicle production affects the mean and Fano factor of the mature peroxisome abundance distribution. *Figure 3—figure supplement 4* depicts the Fano factors of the Golgi apparatus and vacuole abundance distributions from cells grown in oleic acid-rich medium. Error bars are ±1 standard error of the mean, estimated by bootstrapping.

The following figure supplements are available for figure 3:

**Figure supplement 1**. Incorporating pre-peroxisomal vesicle fusion into the model of de novo peroxisome biogenesis.

**Figure supplement 2**. Measuring mature peroxisome abundance statistics using Pex3-mRFP as the peroxisomal marker.

**Figure supplement 3**. Predicting stochastic fluctuations in peroxisome abundance distributions in budding yeast cells synchronized and arrested in S-phase of the cell cycle.

**Figure supplement 4**. Effect of increasing pre-peroxisomal vesicle production in Model 2 on mean and mature peroxisome abundance distribution Fano factors.

**Figure supplement 5**. Golgi and vacuole abundance distribution Fano factors obtained from cells cultured in oleic acid rich medium.

abundance fluctuations for peroxisomes (*Figure 3C*, *Figure 3—figure supplement 2B*), with a measured Fano factor of $\sigma^2/<n> = 2.4 \pm 0.2$ (*Figure 3D*, *Figure 3—figure supplement 2C*). Importantly, when the rates of pre-peroxisomal vesicle production and fusion are increased to match the increased mean mature peroxisome abundance observed in cells cultured in oleic acid medium in the alternative model of peroxisome biogenesis (*Figure 3—figure supplement 4A*), the alternative model cannot explain this rise in the mature, import-competent peroxisome abundance distribution Fano factor upon transfer of cells to oleic acid containing medium (*Figure 3—figure supplement 4B*). Therefore, we conclude that increased pre-peroxisomal vesicle production and fusion cannot explain the rise in the mature peroxisome abundance distribution Fano factor obtained from cells cultured in oleic acid medium. Instead we infer that upon transfer to oleic acid containing medium budding yeast cells primarily generate new peroxisomes by fission of pre-existing peroxisomes.

To test whether our results are specific to peroxisomes or a more general aspect of culturing in oleic acid rich conditions, we also measured Golgi and vacuole abundance distributions in oleic acid rich medium. We confirm that for the Golgi apparatus the Fano factor remains virtually unchanged at $\sigma^2/<n> = 1.0 \pm 0.2$ while for the vacuole the Fano factor increases to $\sigma^2/<n> = 0.8 \pm 0.1$ respectively (*Figure 3—figure supplement 5*). This rise in the vacuole abundance distribution Fano factor is consistent with the increase in mean vacuole abundance in oleic acid cultured cells from $<n> = 2.1$ to $<n> = 4.0$.

## Predicting organelle abundance distributions and Fano factors in diploid cells

Our model of organelle biogenesis makes predictions about how the magnitude of organelle abundance fluctuations will change if the mean organelle abundance is changed. For example, in the case of the Golgi apparatus and peroxisomes in glucose-grown yeast cells, where de novo synthesis and first order decay dominate the organelle abundance distributions, we expect that even if the mean is increased the distribution will still follow a Poisson distribution and the Fano factor will remain at $\sigma^2/<n> = 1$. On the other hand, for the case of the vacuole if the mean vacuole abundance is doubled from 2 to 4, we expect that while the distribution will still follow a shifted Poisson distribution, the Fano factor, which recall follows $\frac{\sigma^2}{\langle n \rangle} = 1 - \frac{1}{\langle n \rangle}$ for the shifted Poisson, will increase from $\sigma^2/<n> = 0.5$ to $\sigma^2/<n> = 0.75$.

We conjectured, motivated by the observation that organelle sizes scale with cellular volume and ploidy (*Weiss et al., 1975*; *Chan and Marshall, 2010*; *Uchida et al., 2011*), that diploid cells may have higher mean numbers of organelles, enabling us to experimentally evaluate our predictions of the relationship between mean and fluctuation size. We constructed diploid versions of the strains used for our above measurements and repeated our imaging experiments on these diploid strains with fluorescently labeled organelles. We found that vacuole and peroxisome abundance distributions have increased means in diploid cells compared to haploid cells; the Golgi apparatus and late Golgi notably did not yield a statistically significant increase in mean abundance in diploid cells (*Figure 4—figure supplements 1, 2*).

In the case of the diploid vacuole abundance distribution, $<n> = 3.6$, which represents a 71% increase from the haploid value of 2.1 (*Figure 4A,B*). The diploid vacuole abundance distribution is still well described by a shifted Poisson distribution that is calculated from the experimentally measured mean, again with no fitting parameters (*Figure 4A*, blue line). Our equation for the Fano factor predicts that when the vacuole mean is $<n> = 3.6$, the Fano factor $\frac{\sigma^2}{\langle n \rangle} = 1 - \frac{1}{\langle n \rangle} = 0.7$, higher than the prediction in the haploid case of $\sigma^2/<n> = 0.5$. In diploid cells expressing the Vph1-GFP marker we measure a Fano factor of $\sigma^2/<n> = 0.7 \pm 0.1$, significantly higher than the haploid value of $\sigma^2/<n> = 0.5 \pm 0.1$ (*Figure 4C*), and consistent with our theoretical prediction.

In the case of peroxisomes, the diploid organelle abundance distributions yield $<n> = 6.8$, respectively, when cells are cultured in glucose-rich medium, representing roughly a doubling in mean abundance in each case compared to the haploid abundance distributions (*Figure 4D,E*). As mentioned above, the organelle abundance distribution should follow the Poisson distribution, even with the higher mean, because the underlying biogenesis mechanisms of de novo synthesis and decay should not have changed. We calculated the Poisson distribution derived from the experimentally measured mean abundances and find that the predicted distributions, even with no fitting parameters, closely describe the experimentally measured abundance distributions (*Figure 4D*). Finally, the Fano factor for the

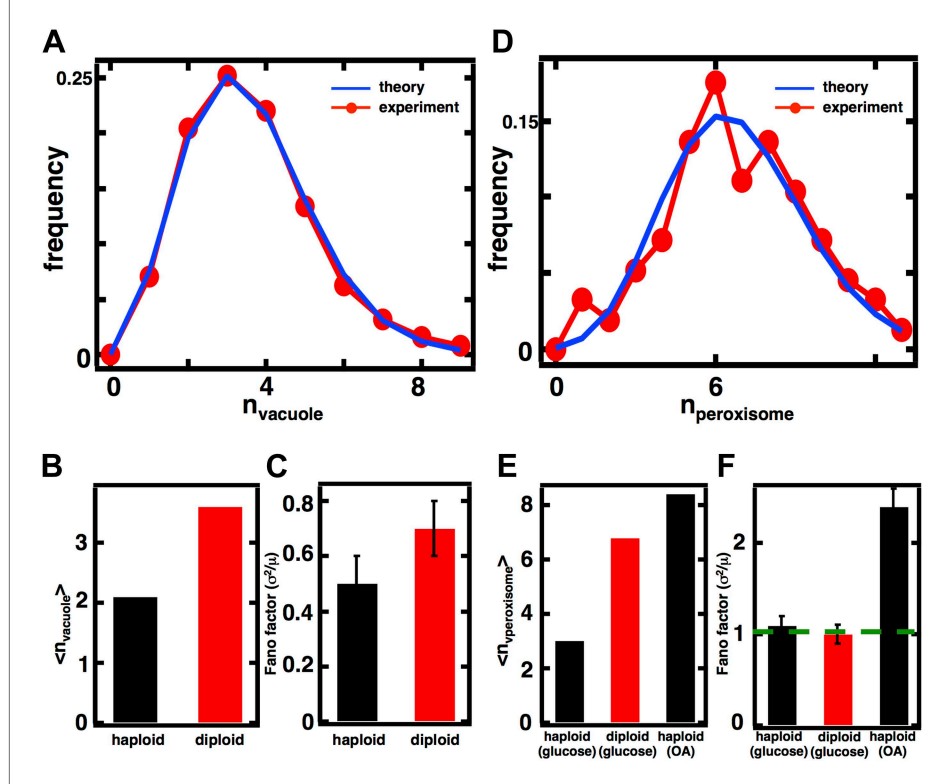

**Figure 4**. Predicting organelle abundance fluctuations in diploid cells. (**A**) Histograms depicting the theoretically predicted vacuole abundance distribution (blue trace) and experimentally measured single diploid cell vacuole abundance distribution (red trace). N = 127 cells were analyzed. (**B** and **C**) Bar charts comparing experimentally measured vacuole abundance means (**B**) and Fano factors (**C**) in haploid and diploid cells cultured in glucose-rich medium. (**D**) Histograms depicting the theoretically predicted peroxisome distributions (blue trace) and experimentally measured single diploid cell peroxisome abundance distributions (red trace). N = 154 cells were analyzed. (**E** and **F**) Bar charts comparing experimentally measured peroxisome means (**E**) and Fano factors (**F**) in haploid and diploid cells cultured in glucose-rich or oleic acid (OA) medium. Green line in panel (**F**) indicate Fano factor of 1. Error bars are ±1 standard error of the mean, estimated by bootstrapping. *Figure 4—figure supplement 1* displays the Golgi apparatus abundance distribution and its Fano factor from diploid cells grown in glucose-rich medium. *Figure 4—figure supplement 2* depicts data similar to panels (**D**–**F**) but with Pex3-mRFP as the peroxisome marker.

The following figure supplements are available for figure 4:

**Figure supplement 1**. Golgi apparatus abundance fluctuations in diploid cells.

**Figure supplement 2**. Late Golgi apparatus abundance fluctuations in diploid cells.

**Figure supplement 3**. Peroxisome abundance fluctuations in diploid cells measured by Pex3-mRFP.

peroxisome abundance distribution should remain $\sigma^2/\langle n\rangle = 1$; a hallmark of the Poisson distribution is that the Fano factor is 1 for any mean, a fact also reflected in *Equation 1* with $k_{fusion} = k_{fission} = 0$. The measured Fano factor matches the theoretical prediction closely at $\sigma^2/\langle n\rangle = 1.0 \pm 0.1$ for the peroxisome (*Figure 4F*). As before, we obtain indistinguishable results for peroxisomes labeled by fusing the peroxisomal membrane protein Pex3 to mRFP (*Figure 4—figure supplement 3*).

Interestingly, the Fano factors for the peroxisome abundance distribution in diploid cells grown in glucose are much smaller than the Fano factors measured in cells grown in oleic acid-rich medium (*Figure 4F*) despite the fact that their mean values are similar (*Figure 4E*). This provides additional evidence that peroxisome biogenesis involves fundamentally different mechanisms in glucose-cultured

cells vs oleic acid-cultured cells, with our model pointing to the less noisy de novo synthesis pathway dominating the former and the more noisy fission pathway dominating the latter.

## Deleting peroxisome fission factors restores abundance distributions toward the Poisson limit

Finally, we tested the mechanistic prediction that peroxisomes switch from de novo synthesis dominated production in glucose-containing medium to fission dominated production in oleic acid-containing medium. Peroxisome fission is mediated by the proteins Vps1, Dnm1 and its accessory protein Fis1, with the dominant role played by Vps1 (*Kuravi et al., 2006*). We engineered yeast strains containing fluorescently labeled peroxisomes and lacking either Dnm1, Vps1, or Fis1, cultured these cells in oleic acid medium, counted the number of peroxisomes in each cell, and calculated the mean and Fano factor of the peroxisome abundance distributions for each strain (*Figure 5*). We also repeated these experiments in yeast cells bearing peroxisomes labeled by fusing Pex3 with mRFP and obtained virtually indistinguishable results (*Figure 5—figure supplement 1*). We expect that if fission dominates peroxisome proliferation in cells cultured in oleic acid medium, the mean peroxisome abundance will decrease upon deletion of Vps1, Dnm1 and Fis1, and the Fano factor for the peroxisome abundance distribution for the mutant strains will decrease toward $\sigma^2/<n> = 1$, the value for de novo synthesis dominated production. We find that the mean peroxisome abundance decreases by fourfold in *vps1Δ* cells, while the peroxisome abundance distribution Fano factor is reduced threefold. As expected, the effects in *fis1Δ* and *dnm1Δ* cells are weaker: the mean peroxisome abundance decreases by slightly less than twofold in *fis1Δ* cells and roughly 10% in *dnm1Δ* cells, while the Fano factors for *fis1Δ* and *dnm1Δ* cells are reduced by ~30% and ~15% respectively. Most importantly, the Fano factor measured for *vps1Δ* cells is $\sigma^2/<n> = 1.0$, exactly what one would expect for de novo synthesis dominated peroxisome production. These results strongly suggest that fission dominates peroxisome proliferation in wild-type cells cultured in oleic acid medium.

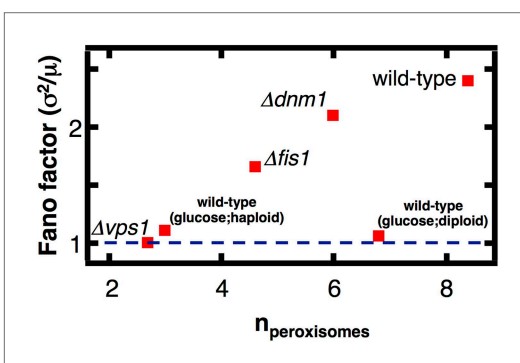

**Figure 5**. The effect of fission factor deletions on peroxisome abundance statistics. Budding yeast cells containing peroxisomes labeled by YFP-PTS1 and lacking one of the peroxisome fission factors *VPS1*, *DNM1* or its accessory factor *FIS1* were cultured in medium containing 0.2% oleic acid for 20 hr. Single cell peroxisome abundance distributions were measured for each of these strains. The Fano factors are plotted as a function of mean peroxisome abundance extracted from the single cell peroxisome distributions. *Figure 5— figure supplement 1* shows data similar to *Figure 5* but with Pex3-mRFP as the peroxisome marker.

The following figure supplements are available for figure 5:

**Figure supplement 1**. Peroxisome abundance fluctuations in cells containing deletions of the fission factors Dnm1, Fis1 and Vps1 measured labeling of peroxisomes by Pex3-mRFP.

## Discussion

Here we formulate a stochastic model of organelle biogenesis and find that the diverse mechanisms that alter organelle abundance leave distinct signatures on the shape of the organelle abundance distribution. Specifically we find that fission or fusion dominated organelle abundance dynamics, respectively, increase or decrease the width of the organelle abundance distribution, as measured by the variance of the distribution divided by the mean (termed the Fano factor), compared to de novo synthesis dominated organelle biogenesis. We then applied this theory to predict the precision with which single budding yeast cells regulate their abundances of the Golgi apparatus and vacuoles. Our results show that budding yeast cells tolerate the theoretically predicted maximum Fano factors consistent with the known mechanisms of Golgi apparatus and vacuole biogenesis. Having validated that the theory could make quantitatively accurate predictions, we then used it to quantitatively distinguish between de novo synthesis-based and fission-based models of peroxisome biogenesis. Our results showed that the peroxisome abundance distribution is consistent with a model in which the organelle is created primarily by de novo synthesis in glucose-cultured budding yeast cells, but then switch to a noisy, fission-dominated biogenesis when cells are cultured in a fatty acid rich environment in

which organelle biogenesis is upregulated. We presented additional lines of evidence for this picture in the form of measuring the fluctuations in peroxisome abundances in mutant yeast strains lacking organelle fission factors. Specifically, consistent with our theoretical prediction, we observed significantly decreased fluctuations in peroxisome abundance when these mutant strains were cultured in fatty acid rich medium compared to our measurements in wild-type cells.

## Endomembrane organelle abundance involves little feedback regulation

Perhaps the most surprising aspect of the comparison between our model of organelle abundances and measurements of endomembrane organelle abundance distributions is the close match between theory and experiment without the need to modify the model to take account feedback control mechanisms. This is in stark contrast to the expectation that cells know how to count and tightly control the number of its various organelles (*Rafelski and Marshall, 2008*). While some subcellular structures such as the nucleus and centrioles are clearly under the control of strong feedback control mechanisms that suppress fluctuations in their abundances (*Marshall, 2007*), the cell apparently tolerates the maximum amount of variability in endomembrane organelle abundance generated by a given set of biogenesis mechanisms. It must be noted, however, that while the close match between our theory and experiment without invoking any feedback control implies that any feedback control mechanisms operating on endomembrane organelle copy numbers are no more precise than the biogenesis system could achieve without feedback, we do not explicitly rule out the presence of feedback regulation of the organelle copy numbers. Our model is also very simple, and though we were able to obtain new insights into organelle biogenesis using the model we are likely suppressing details that could affect organelle abundance under certain conditions. For example, it has been recently shown that the rate of peroxisome decay via autophagy depends on the existence of a functional fission pathway (*Mao et al., 2014*). Nevertheless, it would be trivial to extend our model to incorporate such findings; in the case of fission-dependent autophagy rates, for example, one would simply need to rewrite the decay rate to be a function of the fission rate rather than just being a simple numerical parameter. Such effects would of course require us to take greater care in interpreting the results of the model, but do not invalidate that model; indeed deviations from our simple model could even facilitate discovery of effects such as coupling between the different processes affecting organelle copy number and feedback control of organelle copy numbers. Finally, the consequences of these fluctuations in organelle abundance on the fundamental cell biological processes controlled by these organelles, ranging from secretion to metabolism, remain to be explored. Assessing the dependence of cell biological processes on the abundance of the organelles governing these processes requires systematic, quantitative measurements such as, for example, how the rate of secretion depends on Golgi apparatus abundance or how protein degradation rates depend on vacuole abundances.

## Switch of peroxisome biogenesis to fission suggests rapid response to environmental change

Our results suggest a resolution to the long-running debate over whether peroxisome biogenesis is the result of de novo synthesis or fission (*Hoepfner et al., 2005*; *Motley and Hettema, 2007*; *Mast et al., 2010*; *van der Zand et al., 2012*), with our theory and measurements supporting a model in which peroxisomes are created de novo in glucose-rich conditions, but switch to primarily fission-based proliferation in fatty acid rich environments that demand peroxisomal function.

With these quantitative tools in hand to characterize organelle abundance processes, it will be of great interest to uncover functional reasons why the cell employs de novo synthesis vs fission to proliferate organelles. Given that peroxisomes appear to switch from de novo synthesis to a noisy, fission dominated creation, it will be particularly interesting to measure the degree to which the abundances of peroxisomes with other metabolic organelles such as the mitochondria or lipid droplets are correlated in single cells. These correlation measurements can allow us to infer the design principles underlying cellular responses to fatty acid rich conditions: anti-correlations in noisy peroxisome and lipid droplet production, for example, would suggest a model in which different cells specialize in lipid metabolism vs storage, while correlated production would favor a model in which only a subset of cells specialize in responding to the environmental change. Along with previously developed frameworks examining variability in organelle number (*Hennis and Birky, 1984*; *Marshall, 2007*), our model can aid in examining the functional consequences of stochastic fluctuations in organelle abundance.

## Diverse biological systems are described by same opposing forces as endomembrane organelles

Perhaps most importantly, the generality of our approach makes it amenable to analyzing the wealth of subcellular compartments and granules in prokaryotes (*Yeates et al., 2008*) and even eukaryotes whose biogenesis mechanisms we do not yet understand or are just discovering (*Narayanaswamy et al., 2008*). With previous examinations of organelle number control (*Hennis and Birky, 1984*; *Marshall 2007*) and our analysis of the peroxisome fission pathway as guides, we hope that our model will be used as a framework in which to interpret future genetic studies that aim to uncover the biophysical pathways responsible for the biogenesis of subcellular structures.

## Materials and methods

### Strains

All strains were taken from the collection of GFP fusion strains generated by *Huh et al. (2003)*. This collection includes one set of strains used as organelle references against which the localization of all other fluorescently tagged proteins were scored. Strains from this reference strain set contain the monomeric red fluorescent protein (mRFP) fused to a protein that localizes to a specific organelle with high reliability. Where possible, as was the case for endosomes, Golgi apparatus, and peroxisomes, we selected these organelle reference strains for visualization. In order to visualize mature, import-competent peroxisomes, we engineered a yeast strain that expressed a fusion of the peroxisome targeting signal 1 (PTS1), consisting of the amino acids serine, lysine and leucine, to the extreme C terminus of the monomeric, yeast-enhanced Citrine fluorescent protein (yemCitrine). We used fusion PCR to generate a construct consisting of the yemCitrine-PTS1 gene flanked by the promoter of the *TEF2* gene on the 5′ end and the transcriptional terminator element of the *ADH1* gene on the 3′ end and integrated this construct into the yeast genome at the *HIS3* locus. For the case of the vacuole, for which no mRFP reference strain was created, we used a strain from the library that was engineered to contain the green fluorescent protein (GFP) fused to the vacuolar membrane protein Vph1.

### Culture conditions

For glucose medium, strains were grown to mid-log phase at 30°C in standard synthetic medium containing 2% glucose and subsequently imaged. For oleic acid medium, strains were grown to mid-log phase at 30°C in standard synthetic medium containing 2% glucose, washed twice, and resuspended in medium containing 0.3% yeast extract, 0.6% peptone, 0.1% glucose, 0.1% Tween40, and 0.2% oleic acid, cultured for 20 hr in the oleic acid rich medium, and subsequently imaged.

### S-phase arrest

Mating type **a** (MAT**a**) yeast cells expressing green fluorescent protein (GFP) labeled organelles from the *Huh et al. (2003)* collection were grown to $OD_{600}$ = 0.1 in complete synthetic medium. 10 µM alpha factor were then added to the cell culture for 2 hr to arrest the cells in the G1 phase of the cell cycle. Following washing of the cells twice with 50 ml complete synthetic medium without alpha factor to remove the alpha factor from the culture medium, the G1 synchronized cells were transferred into medium containing 100 mM hydroxyurea to arrest the cells in S-phase. Following 2 hr of hydroxyurea treatment, the cells were fixed with 3.7% formaldehyde, washed, and imaged.

### Imaging conditions

Between N = 100–300 cells were imaged on an inverted Olympus IX-71 microscope fitted with a Perkin–Elmer spinning disc including a Yokogawa head. Samples were illuminated with laser light at 561 nm (mRFP) and 491 nm (GFP) and imaged on a Hammamatsu EMCCD camera.

### Image processing

Cell segmentation was performed manually using ImageJ. To obtain the number of organelles in each cell, we split the data into two cases. For those organelles that appear as discrete foci (endosomes, Golgi apparatus, peroxisomes) individual slices of each image stack were filtered with a Gaussian blurring filter to eliminate high frequency noise in the image, followed by a Laplacian second derivative filter to sharpen edges thereby enhancing the foci. The filtered images were then thresholded to identify those pixels belonging to an organelle. Finally the organelles could be assigned to single cells using the manually segmented image. This analysis was carried out in MATLAB. All organelle

identification and assignment to single cells was manually verified. In the second case, where the organelle (vacuole) appears as a discrete ring, all quantification of number of organelles per cell was done manually.

## Acknowledgements

We thank D Huh and J Choi for fruitful discussions, Q Justman for assistance with the cell cycle arrest experiments, and WF Marshall, BM Stern, and V Denic for critically reading the manuscript. This work was supported by the Howard Hughes Medical Institute (EKO) and a Ruth L Kirchstein National Research Service Award (F32GM101751) from the National Institutes of Health (SM).

## Additional information

### Competing interests

EKO: Erin O'Shea is Chief Scientific Officer and a Vice President at the Howard Hughes Medical Institute, one of the three founding funders of *eLife*. The other author declares that no competing interests exist.

### Funding

| Funder | Grant reference number | Author |
| --- | --- | --- |
| Howard Hughes Medical Institute | | Erin K O'Shea |
| National Institutes of Health | F32GM101751 | Shankar Mukherji |

The funders had no role in study design, data collection and interpretation, or the decision to submit the work for publication.

### Author contributions

SM, Conception and design, Acquisition of data, Analysis and interpretation of data, Drafting or revising the article; EKO, Conception and design, Drafting or revising the article

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
