## [Decision Letter]

Thank you for sending your work entitled “Mechanisms of organelle biogenesis shape stochastic fluctuations in organelle abundance” for consideration at *eLife.* Your article has been favorably evaluated by a Senior editor and 3 reviewers, one of whom, Vivek Malhotra, is a member of our Board of Reviewing Editors.

The Reviewing editor and the other reviewers discussed their comments before we reached this decision, and the Reviewing editor has assembled the following comments to help you prepare a revised submission.

1) In the model, organelle abundances arise from a combination of creation (by whatever means) and first-order decay. The authors claim that the decay term represents “an aggregation of a number of processes such as cell division, heterotypic fusion or autophagy”. In a fissioning cell population for high abundances, dilution can be approximated by a first-order decay term; but this is not true for low abundances. Furthermore, non-trivial behaviour can arise if partitioning between parent and daughter cells is not binomial. The authors would have to show (by simulation or any other means) why an explicit incorporation of partitioning during cell division is not required in their model. It could be, for example, that in a budding organism the daughter does not significantly affect organelle properties in the mother, and therefore the authors' model is a reasonable approximation.

2) There will be cell cycle variation in any of the rates considered; this will appear as an “extrinsic” term in the coefficient of variation, or equivalently a term proportional to the mean in the Fano factor. I assume from the Methods section that the authors pool all cells together (i.e., use a mixed or asynchronous culture). Using images alone, it is possible for the data to be partitioned by cell-cycle phase. Does this reveal identical organelle number distributions for all phases? If not, how much variation does one see? Another way to experimentally estimate the size of the extrinsic term is to see the change in Fano factor as a function of the mean. The authors have shown how this can be achieved by a shift to oleic acid medium or by using diploid cells. Can they use the resulting measurements to bind the extrinsic term?

3) The extrinsic noise issue is relevant to the peroxisome data. Here, the authors infer, from obtaining a Fano factor that exceeds 1, that peroxisomes are generated primarily by fission in oleic acid medium. Since the organelle abundance data are from a mixed asynchronous population, it is important to establish that the increase in Fano factor is not due to a cell-cycle-dependent de-novo synthesis rate.

The cell cycle specific issues could be addressed more clearly – experimentally – by arresting cells in a specific stage, S-phase, for example, and then re testing the model.

4) Test if the late Golgi marked with Sec7 behaves the same way as the early Golgi. Either outcome is likely to be highly significant and should be tested experimentally.

---

## [Author Response]

*1) In the model, organelle abundances arise from a combination of creation (by whatever means) and first-order decay. The authors claim that the decay term represents “an aggregation of a number of processes such as cell division, heterotypic fusion or autophagy”. In a fissioning cell population for high abundances, dilution can be approximated by a first-order decay term; but this is not true for low abundances. Furthermore, non-trivial behaviour can arise if partitioning between parent and daughter cells is not binomial. The authors would have to show (by simulation or any other means) why an explicit incorporation of partitioning during cell division is not required in their model. It could be, for example, that in a budding organism the daughter does not significantly affect organelle properties in the mother, and therefore the authors' model is a reasonable approximation*.

We appreciate the reviewers’ concern about how partitioning of organelles between parent and daughter cells can affect our results. While, as the reviewers note below, the Golgi primarily decays through maturation (Losev, et al. Nature 2006), and thus partitioning is not expected to significantly affect our results, vacuole (Catlett and Weisman, Curr Opin Cell Biol 2000) and peroxisome (Fagaransu et al., Nat Rev Mol Cell Biol 2010) copy numbers in parent cells can be significantly impacted by loss via inheritance to daughter cells. Below we will show that both simulation results and previous findings from the literature support the idea that our model is a reasonable approximation of the situation in growing and dividing cells.

First we will detail the results of our simulations to probe how well our model, which we will call Model 1, approximates two alternative models, which we will call Models 2 and 3, that explicitly include partitioning between parent and daughter cells. In Model 2, we replace the first order decay term with a term that reduces the organelle copy number by half in regular intervals, thereby mimicking organelle partitioning upon cell division. Reducing the organelle copy number by half upon each division of a parent cell into 2 daughter cells is necessary for all cells in the population to achieve steady state organelle copy numbers. Furthermore, studies of vacuole and peroxisome partitioning upon cell division that indicate that partitioning occurs much more equitably than expected from a random binomial process (Weisman et al., J Cell Biol 1987; Fagaransu et al., Biochim Biophys Acta 2006), motivating the approximation of partitioning by reducing organelle copy number by exactly one half. In Model 3, we replace the first order decay term with a term that partitions organelles into the daughter cells with probability p = 1/2, mimicking binomial partitioning upon cell division. Finally, to simplify our analysis, we focus our simulations only on the case where de novo synthesis takes place, setting k_fission_ = k_fusion_ = 0. We then ran the simulations of Model 2 and Model 3 10,000 times, built up the organelle copy number distributions resulting from each model and plot the Fano factors of these distributions below.Author response image 1.Comparing Fano factors obtained from simulations of models with alternative organelle decay processes. Model 1 implements first order decay, Model 2 implements reducing the organelle number by half at defined time intervals, and Model 3 implements reducing organelle number in a given cell by binomial partitioning of the organelles into daughter cells.

We see that when we adjust the de novo synthesis rates in Model 1, Model 2, and Model 3 such that they produce the same mean organelle copy number, we observe no significant differences in the Fano factors obtained from simulations of Model 1, Model 2, and Model 3. This analysis is now included as Figure 1—figure supplement 1.

There are two intuitive ways to understand why our approximation of first order decay works well. First, for the Golgi apparatus and the peroxisome, the mean abundance is high enough (empirically roughly 4 and higher) that the dynamics of organelle loss in our first order decay approximation do not substantially differ quantitatively from the dynamics achieved through partitioning of organelles at a defined time. Second, for the Golgi apparatus and the vacuole, the cell division timescale is slow compared to the timescales of the other processes (de novo synthesis and maturation for the Golgi, fission and fusion for the vacuole) affecting organelle copy number. Thus, these other processes come to a steady state on the time scale of cell division and only a very small fraction of cells we measure will deviate from this steady state due to partitioning at cell division.

*2) There will be cell cycle variation in any of the rates considered; this will appear as an “extrinsic” term in the coefficient of variation, or equivalently a term proportional to the mean in the Fano factor. I assume from the Methods section that the authors pool all cells together (i.e., use a mixed or asynchronous culture). Using images alone, it is possible for the data to be partitioned by cell-cycle phase. Does this reveal identical organelle number distributions for all phases? If not, how much variation does one see? Another way to experimentally estimate the size of the extrinsic term is to see the change in Fano factor as a function of the mean*. *The authors have shown how this can be achieved by a shift to oleic acid medium or by using diploid cells. Can they use the resulting measurements to bind the extrinsic term?*

We agree that it is important to experimentally determine the potential effects of the cell cycle on the organelle copy number distributions. As suggested by the reviewer comments, the size of the extrinsic organelle abundance fluctuations due to the cell cycle can be bounded by our data from diploid cells in the case of the vacuole and peroxisome since these organelles experience a doubling in the mean abundance. In both cases, since our data indicates that there is no increased Fano factor compared to what would be expected in a cell cycle independent model, the extrinsic term can be inferred to be near 0.

Nevertheless, to directly measure the effects of cell cycle variation on the organelle abundance distributions, including the Golgi apparatus, we followed the suggestion of the reviewers below and measured organelle abundance distributions from cells arrested in the S-phase of the cell cycle. We accomplished the S-phase arrest, in brief, by growing mating type a (MATa) yeast cells expressing green fluorescent protein (GFP) labeled organelles to OD_600_ = 0.1 in complete synthetic medium, adding 10µ M alpha factor to the cell culture for 2 hours to arrest the cells in the G1 phase of the cell cycle, washing the cells to remove the alpha factor from the culture medium, and then transferring these G1 synchronized cells into medium containing 100 mM hydroxyurea to arrest the cells in S-phase. For the organelles included in our study, we obtained virtually identical Fano factors in S-phase arrested cells compared to our unsynchronized cultures:Author response image 2.Organelle abundance distributions and their Fano factors measured from budding yeast cells arrested in S phase with hydroxyurea. (A, B) Anp1-GFP, (C, D) Sec7-GFP, (E, F) Pex3-GFP, (G, H) Vph1-GFP.

These data are now included as Figure 2—figure supplement 2 (panels A-D, G-H) and Figure 3—figure supplement 3 (panels E and F).

*3) The extrinsic noise issue is relevant to the peroxisome data. Here, the authors infer, from obtaining a Fano factor that exceeds 1, that peroxisomes are generated primarily by fission in oleic acid medium. Since the organelle abundance data are from a mixed asynchronous population, it is important to establish that the increase in Fano factor is not due to a cell-cycle-dependent de-novo synthesis rate*.

*The cell cycle specific issues could be addressed more clearly – experimentally – by arresting cells in a specific stage, S-phase, for example, and then re testing the model*.

As shown in the figure above (Figure 7, panels E, F), we obtained the same Fano factor for peroxisome distributions measured from a population of yeast cells arrested in S-phase of the cell cycle as we obtained with unsynchronized cultures. However, since peroxisome abundance decays primarily through dilution from cell division, the cell cycle arrest experiment suggested by the referee that we carried out above will not only arrest the cells in a specific cell cycle stage but will also reduce the peroxisome decay rate. The reduced peroxisome decay rate thus increases peroxisome copy numbers in the cell cycle stage the cells are arrested in even if there were no cell cycle dependent de novo synthesis rate at all.

Using the diploid data to bind the extrinsic term due to variability in the cell cycle phase in the population of cells is therefore particularly useful for the case of the peroxisome. If, for example, the de novo peroxisome synthesis rate were twice as large in the G1-phase of the cell cycle as in the other phases, then the measured peroxisome abundance distribution would be the convolution of two Poisson distributions, each parameterized by the cell cycle phase specific de novo synthesis rate. Doubling each cell cycle phase specific de novo synthesis rate would double the mean peroxisome abundance, but also pull apart the two underlying Poisson distributions since the mean of the G1-phase specific distribution would increase by much more in absolute terms than the mean of the non G1-phase distribution; indeed in the most extreme case the resulting peroxisome abundance distribution could even become bimodal, with peaks representing the G1-phase and non-G1-phase specific de novo synthesis constants. The change in the Fano factor upon the increased de novo synthesis rate is thus directly proportional to the ratio of the de novo synthesis rates in G1 phase versus non-G1 phase of the cell cycle. In glucose-containing medium, since we obtain a near-doubling of the mean peroxisome abundance in diploid cells without any accompanying increase in the peroxisome abundance distribution Fano factor, we conclude that peroxisome de novo synthesis is unlikely to occur with a cell-cycle-dependent rate.

Formally, the increase in the Fano factor in oleic acid medium could be due to an oleic acid-specific increase in a cell cycle-dependent de novo synthesis rate. In this case, as described above, the peroxisome abundance distribution measured from asynchronous cultures would be a convolution of two or more Poisson distributions and result in a higher measured Fano factor even in the absence of fission. To test this possibility, we binned our image data taken from cells expressing fluorescently labeled peroxisomes cultured in oleic acid medium into two categories: one category for cells without a bud, which represent cells in the G1-phase of the cell cycle; and another category for cells with either a small or large bud, which represent cells in the S-phase, G2-phase, or M-phase of the cell cycle. If a cell cycle dependent de novo synthesis rate were to explain our data then both categories of cells would exhibit peroxisome abundance distributions with Fano factors σ^2^/<n> = 1, the hallmark of de novo synthesis driven biogenesis, though presumably with different means. If either category exhibited a peroxisome abundance distribution with Fano factor σ^2^/<n> > 1 the data would be inconsistent with purely de novo synthesis driven peroxisome biogenesis. The subsample of our dataset containing unbudded cells (N = 111 cells) gives rise to a peroxisome abundance distribution with Fano factor σ^2^/<n> = 2.2 +/- 0.3 while the subsample containing budded cells (N=33 cells) gives rise to a peroxisome abundance distribution with Fano factor factor σ^2^/<n> = 2.0 +/- 0.4. This data suggests that substantial fission occurs in all phases of the cell cycle and thus we need not invoke a cell cycle dependent de novo synthesis rate to explain the increase in the Fano factor of the peroxisome abundance distribution measured from our asynchronous populations.

*4) Test if the late Golgi marked with Sec7 behaves the same way as the early Golgi. Either outcome is likely to be highly significant and should be tested experimentally*.

We thank the referees for the suggestion to examine the late Golgi abundance distributions. We repeated our organelle counting experiments on a budding yeast strain expressing the Sec7 protein fused to the green fluorescent protein (GFP) taken from the Huh et al. genome-wide GFP-tagged yeast strain library. We would expect that for the late Golgi, the abundance distribution would be governed by a balance between maturation of the early Golgi, which should be described by stochastic de novo synthesis, and loss through further maturation and as well as cell division, which should be described by first order decay. Thus, as in the case of the Anp1-mRFP labeled Golgi apparatus, we would expect that the Sec7-GFP labeled late Golgi abundance distribution would closely follow Poisson statistics with a Fano factor of 1. As shown below, we see exactly this with the experimentally measured late Golgi abundance distribution excellently matched by the Poisson distribution calculated using the experimentally measured late Golgi copy number, thus using no fitting parameters.Author response image 3.Late Golgi abundance distribution (left panel) and its Fano factor (right panel) in haploid budding yeast cells, measured using Sec7-GFP as a late Golgi marker.

For completeness, we also carried out the late Golgi counting experiments in diploid cells as well as with cells arrested in the S-phase of the cell cycle through treatment with hydroxyurea (data shown above in Figure 7, panels C and D). In both cases, we again see excellent agreement with the predictions from our mathematical framework.

For completeness, we also carried out the late Golgi counting experiments in diploid cells as well as with cells arrested in the S-phase of the cell cycle through treatment with hydroxyurea (data shown above in Figure 2, panels C and D). In both cases, we again see excellent agreement with the predictions from our mathematical framework.Author response image 4.Late Golgi abundance distribution (left panel) and its Fano factor (right panel) in diploid budding yeast cells, measured using Sec7-GFP as a late Golgi marker.

We added the following text to describe these results: “Interestingly, when we repeat these measurements for the late Golgi by performing confocal microscopy on a budding yeast strain expressing the green fluorescent protein (GFP) fused to the late Golgi marker protein Sec7, we see virtually identical results with the late Golgi abundance distribution closely matching a Poisson distribution (Figure 2—figure supplement 1) and thus yielding a measured Fano factor of σ^2^/<n> = 1.0 +/- 0.1 (Figure 2—figure supplement 1).”